# Radiologists’ Perspectives on AI Integration in Mammographic Breast Cancer Screening: A Mixed Methods Study

**DOI:** 10.3390/cancers17213491

**Published:** 2025-10-30

**Authors:** Serene Si Ning Goh, Qin Xiang Ng, Felicia Jia Hui Chan, Rachel Sze Jen Goh, Pooja Jagmohan, Shahmir H. Ali, Gerald Choon Huat Koh

**Affiliations:** 1Department of General Surgery, National University Hospital Singapore, Singapore 119077, Singapore; 2Saw Swee Hock School of Public Health, National University of Singapore, 12 Science Drive 2, #10-01, Singapore 117549, Singapore; feli.cjh@nus.edu.sg (F.J.H.C.);; 3NUS Yong Loo Lin School of Medicine, National University of Singapore, 10 Medical Dr, Singapore 117597, Singapore; 4Department of Diagnostic Imaging, National University Hospital Singapore, Singapore 119074, Singapore

**Keywords:** Breast cancer, artificial intelligence, mammography, screening, radiologists’ perspectives, technology adoption

## Abstract

**Simple Summary:**

Radiologists’ views matter for how new tools are used in real clinics. We studied how radiologists in Singapore think about bringing artificial intelligence into mammographic breast cancer screening. Using a survey and in-depth interviews, we found most radiologists welcome artificial intelligence as a helpful aid for routine reading and reducing fatigue, but do not want it to replace human judgment, especially for complex cases. They were most confident in systems that were tested on local patients and supported by clear national guidance. Concerns included false positives that add work, poor fit with current computer systems, and uncertainty about legal responsibility. These insights can guide researchers, developers, and policymakers to design safer, better-integrated tools, set practical rules, and focus future studies on local validation, usability, training, and accountability.

**Abstract:**

Background/Objectives: Artificial intelligence (AI) is increasingly applied in breast imaging, with potential to improve diagnostic accuracy and reduce workload in mammographic breast cancer screening. However, real-world integration of AI into national screening programs remains limited, and little is known about radiologists’ perspectives in Asian settings. This study aimed to explore radiologists’ perceptions of AI adoption in Singapore’s breast screening program, focusing on perceived benefits, barriers, and requirements for safe integration. Methods: We conducted a mixed methods study involving a cross-sectional survey of 17 radiologists with prior experience using AI-assisted mammography, followed by semi-structured interviews with 10 radiologists across all three public healthcare clusters. The survey measured confidence in AI, attitudes toward its diagnostic role, and integration preferences. Interviews were analyzed thematically, guided by the Unified Theory of Acceptance and Use of Technology (UTAUT) framework. Results: Among survey respondents, 64.7% recommended AI as a companion reader, though only 29.4% rated its performance as comparable to humans. Confidence was highest when AI was validated on local datasets (mean 9.3/10). Interviews highlighted AI’s strengths in routine, fatigue-prone tasks, but skepticism for complex cases. Concerns included false positives, workflow inefficiencies, medico-legal accountability, and long-term costs. Radiologists emphasized the importance of national guidelines, local validation, and clear role definition to build trust. Conclusions: Radiologists support AI as an adjunct to, but not a replacement for, human readers in breast cancer screening. Adoption will require robust regulatory frameworks, seamless workflow integration, transparent validation on local data, and structured user training to ensure safe and effective implementation.

## 1. Introduction

Artificial intelligence (AI) is increasingly recognized for its potential in breast imaging, particularly in enhancing diagnostic accuracy and efficiency [1,2]. This is especially relevant for breast cancer, the most common cancer among women globally, with 685,000 deaths reported in 2020 [3]. Early detection improves survival, making effective breast screening essential [4]. In Singapore, the national program employs a double-reader, blinded-read system, where two radiologists independently assess mammograms, with discordances reviewed by a third. While this model is the current gold standard, it can be labor-intensive and exacerbates existing manpower shortages [5].

Mammograms are now stored as digital images, which allows for computational analysis and facilitates integration with AI algorithms for image interpretation. Studies have shown that AI can significantly reduce radiologist workload [6]. The MASAI trial in Sweden demonstrated that AI-assisted screening matched the cancer detection rate of standard double reading while cutting screen-reading workload [7]. In Denmark, AI implementation reduced workload by 33.5% and improved detection rates from 0.70% to 0.82% [8]. Deep learning models also improve sensitivity to subtle findings that may be missed by human readers.

Despite these advantages, AI adoption remains limited. Barriers include legal uncertainty, lack of trust, and unclear regulatory roles for AI, such as triage tools, companion readers, or replacements [9,10]. Studies from the UK [11] and Sweden [12] show that while radiologists welcome AI as a supportive tool, they are hesitant to rely on it entirely. These studies, however, were conducted in high-income Western settings and may not reflect adoption dynamics in Asia. To fill this gap, our study presents the first mixed methods exploration of radiologists’ perspectives on AI in breast cancer screening in the Asian context. Understanding these perspectives is important to addressing adoption barriers and maximizing AI’s potential in regional screening programs.

## 2. Methods

### 2.1. Study Design and Setting

This is a mixed-methods study, comprising a quantitative survey at an academic healthcare institute in Singapore, followed by qualitative interviews, including individual in-depth interviews across all major public health institutes nationwide. Quantitative surveys were triangulated with qualitative data to help explain the challenges and concerns of widespread adoption of AI by radiologists. The quantitative results were collected from May 2023 to January 2024, as part of a larger multi-reader multi-study trial using AI [13], whilst the qualitative study was conducted from August to December 2024.

We employed a convergent parallel mixed-methods design, in which quantitative and qualitative data were collected during overlapping periods and analyzed separately before being integrated through joint display meta-inference. This design was chosen to enable corroboration and complementarity between numerical trends and experiential insights on AI adoption.

This exploratory mixed-methods design was descriptive rather than inferential; no formal hypothesis testing was undertaken, and we sought to examine radiologists’ experiences and attitudes toward AI integration, complementing concurrent performance-validation trials in Singapore.

### 2.2. Quantitative Methodology

For the quantitative component, a questionnaire-based, cross-sectional survey was adapted and expanded from a previous study [11] (see Appendix A). It was designed to assess radiologists’ perceptions of AI diagnostic capabilities, confidence in using AI, access, conflict resolution preferences, the utility of AI features, and the types of evidence required for AI adoption in mammography screening. The questionnaire was administered to all accredited radiologists in NUHS with a minimum experience in reading at least 500 breast mammograms with AI assistance. Experience with at least 500 mammograms using AI assistance was verified by the Principal Investigator, based on institutional records. This level of experience was deemed necessary to provide insights into the integration of AI in clinical practice, aligning with the study’s goal of capturing perspectives from radiologists with substantial hands-on experience. These included residents, consultants and senior consultants who participated in the multi-reader multi-center study. The questionnaire was administered electronically via a secure institutional survey platform. 

As mentioned above, the questionnaire was adapted from de Vries et al. [11] and reviewed by two senior breast radiologists for contextual relevance. Given the small sample and exploratory nature, formal psychometric validation was deemed not feasible and hence not undertaken.

### 2.3. Qualitative Methodology 

For the qualitative interviews, we recruited a purposive sample of ten radiologists with prior experience of reading digital mammograms with the aid of AI, from all three national healthcare clusters, namely National University Health System (NUHS), Singapore Health Services (SHS), and the National Healthcare Group (NHG). Eligibility criteria were: (1) Radiologists accredited by the breast screening program who have any experience in utilizing AI in facilitating the reading of full field digital mammograms, (2) Willingness to participate in the interview with audio recording, and (3) Ability to complete the interview in English. The sample size was determined based on feasibility, constrained by the limited pool of radiologists nationally who have prior experience with AI in mammography. 

Qualitative interviews conducted in this study utilized the UTAUT (Unified Theory of Acceptance and Use of Technology) framework [14] to structure respondents’ perceptions regarding the strengths and weaknesses of artificial intelligence (AI) in breast cancer screening. The qualitative interview guide is available in the Appendix A (see Appendix A). All interviews were conducted on zoom by co-authors F.C. and R.G.S.G. reviewed all transcripts against the audio-recordings to ensure accuracy. Interviews were conducted until thematic saturation was reached. Saturation was determined when no new codes or themes emerged after the 8th interview. Two additional interviews were conducted to confirm thematic saturation. The interviews lasted between 25 and 45 min. Audio recordings of the interviews were transcribed verbatim for analysis.

The analysis followed a deductive coding approach, where a codebook based on the UTAUT framework was developed prior to analyzing the transcripts. The codebook included predefined categories corresponding to the key constructs of the UTAUT model, such as performance expectancy, effort expectancy, social influence, and facilitating conditions. This theoretical framework guided the initial coding process. As the coding progressed, the categories were iteratively refined to capture emergent themes and nuances in the data that were not fully anticipated by the UTAUT model. Two authors (S.S.N.G. and Q.X.N.) independently generated codes in vivo by repeatedly engaging with the dataset. Any discrepancies between coding schemes were discussed and resolved collaboratively to achieve consensus. 

We selected the UTAUT framework because it is a parsimonious model that integrates constructs from eight earlier technology-adoption theories and it captures both behavioral intention and facilitating conditions, which the study team felt were particularly relevant for clinician technology adoption and hence suited to our study context [14].

### 2.4. Triangulation Approach 

Findings from the quantitative survey and qualitative interviews were triangulated, to identify facilitators and barriers of implementation of AI mammography in breast screening programs. To integrate insights from both phases of data collection, we employed a triangulation approach in which the quantitative and qualitative findings were compared and synthesized through meta-inference. The two datasets were first analyzed independently and then brought together through a joint display to compare areas of alignment and divergence. This process allowed us to identify points of confirmation where quantitative and qualitative results converged, areas of expansion where one method provided additional depth or nuance, and areas of discordance that highlighted aspects of AI adoption requiring further investigation. This integrative approach enabled a richer and more comprehensive understanding of the facilitators, barriers, and real-world considerations influencing AI implementation in breast cancer screening.

The mixed-methods study and reporting adhered to the COnsolidated criteria for REporting Qualitative studies (COREQ) [15] and Good Reporting of a Mixed Methods Study (GRAMMS) checklists [16], see Appendix A. To ensure analytic rigor and trustworthiness in alignment with our interpretive epistemology, we employed several strategies throughout the qualitative process. First, reflexive engagement was maintained via iterative discussions among the core analysts to surface assumptions, examine potential biases, and preserve sensitivity to context. Second, a clear audit trail was documented, including codebook revisions and decision points during interpretation, supporting transparency and methodological accountability. Third, constant comparison across transcripts enabled consistency in theme development while remaining open to emergent nuances. We recognize, in line with recent scholarship, that quality is not achieved through procedural checklists alone, but through reflexivity, contextual grounding, and transparency in interpretive work [17]. This approach is also consistent with the study’s use of the UTAUT framework as an interpretive lens for analyzing technology adoption in practice. 

### 2.5. Statistical Analysis

For the quantitative data analysis, continuous variables were summarized as means with standard deviations (SD) for normally distributed data and medians with interquartile ranges (IQR) for non-normally distributed data. Categorical data are presented as frequencies and percentages (%). The analysis was performed in SPSS v29.

### 2.6. Ethical Approval

The study was approved by the Department Ethics Review Committee of the National University of Singapore (approval number SSHSPH-274). All participants received a study information sheet and provided written informed consent prior to participation. For the interview component, explicit consent was also obtained for audio-recording, transcription, and the use of de-identified quotations in research outputs. Participation was voluntary, and respondents could withdraw at any time without penalty.

## 3. Results

### 3.1. Overview of Findings

All 17 radiologists from the larger multi-reader trial completed the quantitative survey (100% response rate within this cohort). The sample comprised 9 junior residents, 4 senior residents, and 4 consultants from a single department, each with at least 500 AI-assisted mammographic reads. AI was used as a companion, not a second reader, with radiologists reviewing AI output alongside their own interpretations. Ten radiologists participated in the qualitative interviews (mean age: 39.7 years, SD 7.2; 80% male). Median radiology experience was 14 years (IQR 6–16), and AI exposure ranged from 6 to 18 months. Participants came from all three national healthcare clusters: NUHS (*n* = 6), SHS (*n* = 3), and NHG (*n* = 1). All had used FxMammo or Lunit INSIGHT MMG. Thematic saturation was reached after 8 interviews. To confirm saturation, two additional interviews were conducted, which further supported that no new themes emerged. A joint display of the meta-inference can be reviewed in Table 1 and the findings are further elaborated upon below.

### 3.2. Triangulation of Findings

#### 3.2.1. AI’s Role in Supporting Diagnostic Tasks and Ensuring Consistency

Quantitative Findings: From the survey, 29.4% (5/17) of radiologists rated AI’s diagnostic capabilities as comparable to human radiologists, while 70.6% (12/17) felt AI performed worse. A total of 76.5% (13/17) of radiologists stated that AI should not replace human radiologists entirely but 64.7% (11/17) saw AI as a useful companion reader while 41.2% (7/17) supported AI for triage (Figure 1).

Qualitative Findings: Radiologists saw AI as helpful for routine tasks and fatigue-prone scenarios. Participant 1 (P1) noted AI helps “cover the corners”. P2, P9 highlighted AI’s ability to reduce human error, especially in scenarios of radiologist fatigue. P4 shared “An ideal situation will be if AI can triage screening mammograms, prioritizing suspicious cases for human reading first”, supporting the survey’s findings of AI’s potential to augment but not replace human decision-making. 

Triangulation Insight: Both quantitative and qualitative data support AI’s role as a supplementary aid rather than a replacement, with value placed on its consistency in simpler tasks.

#### 3.2.2. Integration and Functional Alignment of AI in Clinical Workflows

Quantitative Findings: When asked about which reader(s) should be given access to use of AI as a companion reader in the double-reader workflow, 82.4% (14/17) chose the first reader, 70.6% chose the second reader (12/17) and 35.3% (6/17) chose the referee/arbitration panel. In discordant cases, 52.9% (*n* = 9) of radiologists favored prioritizing human judgment, while 47.1% (8/17) advocated for resolving disagreements through discussions amongst radiologists (Figure 2).

Qualitative Findings: Interview participants discussed concerns about false positives and workflow integration issues. P4 shared, “It will highlight something, but there’s no corresponding abnormality in another view, and it turns out to be normal in the end,” pointing out AI’s propensity for false positives. P6 highlighted “AI is not good at integrating patient information to make more informed decisions”. P9 and P10 provided examples “AI cannot differentiate (between) post-operative and post-radiation changes from cancers in the breast, (and) flags these as abnormal”, “AI cannot compare current mammograms with prior ones, (and) this leads to decreased accuracy in re-screening”. P1 remarked, “It cannot require too many clicks… otherwise, I’ll forgo using it entirely,” reflecting concerns about workflow inefficiencies. 

Triangulation Insight: AI’s diagnostic and interface limitations reduce workflow efficiency. Radiologists favor maintaining human control in critical decisions. 

#### 3.2.3. Attitudes Toward AI Adoption 

Quantitative Findings: Regarding AI adoption, 76.5% (13/17) of respondents felt that AI cannot replace human radiologists, though 23.5% were open to the possibility in the future. Radiologists’ confidence in using AI as a companion reader was moderate, with a mean score of 5.4 out of 10 (SD = 1.8). Confidence in making correct diagnoses with AI assistance was higher, with a mean score of 6.4 out of 10 (SD = 2.0) (Figure 3). 

Qualitative Findings: Radiologists expressed cautious optimism but stressed the need for more evidence before trusting AI for primary diagnostic tasks. In terms of using AI as a companion reader, P8 exclaimed “I am quite happy to use AI as a second sieve.” P7 shared, “AI can speed things up… so I can spend more time reading advanced scans instead.” In the context of replacing a reader, P2 stated, “I haven’t seen enough evidence that it’s accurate enough to replace humans,” while P5 added, “AI is not sophisticated enough to replace humans yet, but possible in the future.” P7 expressed concern that over-reliance on AI might erode their confidence in personal judgment.

Triangulation Insight: Radiologists show moderate confidence in AI as a support tool but remain hesitant to adopt it as a primary diagnostic tool due to trust issues, personal anxieties, and limited evidence.

#### 3.2.4. Ethical and Regulatory Concerns

Quantitative Findings: Confidence in AI was highest when the AI software had been tested with local data (mean = 9.3/10, SD = 1.0). Confidence in AI was higher when its use was aligned with established national guidelines for breast cancer screening guidelines (mean = 8.8/10, SD = 1.4). However, confidence was lowest in vendor-provided internal analyses (mean = 5.8/10, SD = 3.2), suggesting a lack of trust in commercially provided data (Figure 4). 

Qualitative Findings: Ethical concerns, particularly around medico-legal issues, were prominent in the interviews. P10 highlighted, “Whenever we start using AI, insurance and legal coverage will have to increase,” while P4 questioned, “Who’s going to be responsible if the AI fails?”. P3 emphasized the need for governmental oversight, “If it’s going to be a nationwide implementation, the government needs to step in and set disclaimers about AI’s role.” These concerns echo the low confidence in internationally marketed vendor-driven AI validation seen in the survey results, as radiologists prefer independent, locally tested AI models. 

Triangulation Insight: Both datasets highlight trust as a key issue; radiologists prefer AI validated locally over vendor data, and call for clear ethical and regulatory frameworks to address medico-legal concerns. 

#### 3.2.5. Unintended Consequences and Paradoxical Outcomes

Quantitative Findings: Only 41.2% (7/17) agreed that AI reduces workload, whilst the rest of the respondents rated increased workload due to increased time deliberating on discordant cases. 

Qualitative Findings: Interviewees reinforced concerns that AI may paradoxically increase rather than reduce workload. P9 shared, “AI has generated more work because some cases I thought were normal, but the AI flags something, I feel obliged to investigate more.” Similarly, P3 said, “AI as second reader in Breast Screen Singapore may actually result in longer reads and more discussion of discordant cases.” These insights suggest that AI’s tendency to flag non-actionable findings adds tasks rather than streamlining work. P10 also warned of financial risks: “AI companies may start by offering a cheap deal… Once you’re reliant on it, they raise the price… you have to keep on paying,” raising concerns about hidden long-term costs. 

Triangulation Insight: The paradoxical increase in workload due to AI awas highlighted. Although AI holds promise in reducing time spent on routine cases, its tendency to generate false positives can lead to additional work and cognitive load.

## 4. Discussion

The rapid proliferation of AI technologies in healthcare has transformed diagnostic, prognostic, and workflow paradigms across multiple clinical domains [18]. This mixed methods study found that while 64.7% of radiologists support AI as a companion reader in breast cancer screening, only 29.4% considered its performance comparable to human radiologists. Confidence was highest when AI was tested using local data (mean 9.3/10). The emergent themes from the interviews also align closely with the UTAUT constructs: Performance Expectancy corresponded to perceptions of AI’s accuracy and consistency (Theme 1); Effort Expectancy related to workflow fit and usability (Theme 2); Social Influence encompassed institutional culture and peer expectations (Theme 3); and Facilitating Conditions included regulatory clarity and local validation (Theme 4). This mapping underscores how the UTAUT model provides a coherent theoretical lens for understanding clinicians’ adoption behaviors (thematic concept map shown in Figure 5).

Majority (64.7%) supported AI assistance in the screening process, echoing previous findings that radiologists believe AI can reduce interpretation errors and enhance detection performance [12,19]. Visual aids such as heat maps or bounding boxes were especially valued for improving interpretive accuracy. Despite recognizing these benefits, radiologists emphasized that AI should support rather than replace human expertise. Radiologists’ hesitation for AI to take on primary diagnostic responsibility is consistent with recent work emphasizing that clinicians often hold AI to higher performance standards than humans, particularly due to fears of medico-legal liability and diminished trust in automated decisions [20]. Moreover, radiology work is embedded in tacit knowledge, contextual interpretation, and collaborative decision-making that can be disrupted by tools optimized only for technical performance. AI must therefore be designed to support, rather than constrain, radiologists’ “vision work,” through features that enhance situated judgment and reduce rather than increase cognitive burden [21]. Their concerns mirror findings by Suchman et al., who noted that AI systems lack the contextual reasoning necessary for complex clinical decision-making [22]. Radiologists in our study expressed a strong desire for more robust, peer-reviewed evidence to build trust, consistent with Kelly et al.’s assertion that high-quality evidence is key to adoption [23]. Only 29.4% viewed AI as comparable to radiologists, underscoring AI’s current role as a tool for specific tasks rather than holistic diagnostic decision-making.

While radiologists are generally more open toward AI than other professionals, they also harbor unique apprehensions that AI may challenge their core expertise or allow other specialties to encroach on traditionally radiological roles [24]. Such sociocultural dynamics influence behavioral intention and must be considered alongside technical performance in AI implementation. Radiologists reported technology anxiety and skepticism around algorithmic transparency, issues aligned with the concerns raised by Hanna et al. about ethical implementation requiring clear algorithmic processes [25]. Many described unease when AI contradicted their assessments, leading to cognitive dissonance and hesitation in overriding AI outputs. Some felt that AI developers and radiologists had competing interests, contributing to mistrust [26]. Another barrier was concern over long-term cost unpredictability. Radiologists feared software companies might increase prices after widespread adoption, echoing recent trends in AI pricing models [27]. Thus, effort expectancy was influenced not only by usability but also by emotional and financial stressors related to AI adoption.

Participants highlighted the need for better technical support and user-friendly integration. Poor user interfaces and lack of seamless interoperability with existing systems remain major challenges. The “last mile problem,” where AI tools are not fully integrated into hospital infrastructure, hinders usability [28]. Contrary to expectations, AI sometimes increased workload due to false positives and the need to double-check AI outputs. These issues were especially problematic in patients with dense breasts or post-treatment changes, consistent with findings by Nguyen et al., who identified higher false-positive rates among older patients, Black women, and those with dense breasts [29]. Additionally, ambiguity about legal responsibility for AI-related errors persisted. As noted by Morgan et al., accountability must span from algorithm design to clinical deployment [30].

Notably, while 58.8% of participants in our study perceived that AI may increase workload due to the need to reconcile discordant findings (perceived risk), qualitative interviews confirmed this was already occurring in practice, with several radiologists describing additional cognitive and operational burden when AI flagged clinically non-actionable abnormalities (real risk). Our triangulated analysis demonstrates that radiologists’ concerns are grounded in concrete observations. What begins as a perceived risk (increased workload due to false-positive flags) is experienced as additional image review, supplementary investigations, and prolonged decision-making in discordant cases. This strongly emphasizes that the introduction of AI requires careful workflow redesign to ensure efficiency gains are realized rather than reversed.

Rather than direct peer pressure, social influence manifested through institutional culture. Radiologists expressed ambivalence, acknowledging AI’s inevitability while remaining cautious about full-scale integration. Supportive environments fostering innovation significantly shaped willingness to adopt AI. This mirrors findings by Lee et al., who found that organizational support affects perceived utility and adoption intent [31]. While AI is accepted for narrow tasks, ongoing concerns about its application in complex clinical decisions may limit broader uptake unless transparency, reliability, and usability improve.

Similarly, insights from Ce et al. suggest both junior and senior radiologists generally hold favorable views of AI [32]. However, AI literacy gaps among early-career clinicians and concerns about AI’s diagnostic reliability point to the need for structured education. Chen et al. found that factors such as stress, perceived usefulness, and workload influenced residents’ acceptance of AI, further highlighting the importance of targeted training [33]. Residency programs should incorporate AI curricula encompassing algorithm appraisal, ethical considerations, and practical hands-on exposure. Longitudinal studies are needed to assess how such training affects diagnostic accuracy, clinician confidence, and sustained adoption.

Arguably, the implications of these findings differ across stakeholder groups. For radiologists, the results emphasize the need for structured training programs and workflow integration strategies that enhance confidence in AI use while mitigating cognitive load and over-reliance. For developers, the emphasis lies in adopting human-centered design principles and ensuring rigorous validation of algorithms on local datasets to improve relevance, trust, and performance in specific population contexts. For policymakers and regulators, the study reinforces the importance of establishing clear medico-legal frameworks, transparent accountability structures, and harmonized national guidelines that define the clinical role of AI in breast cancer screening. All in all, stakeholder-specific strategies are critical for achieving safe, equitable, and sustainable integration of AI into real-world radiology practice.

Nonetheless, this study has several notable strengths and limitations. To the best of our knowledge, this is the first mixed methods study on radiologists’ perceptions of AI in breast screening in Asia, addressing a critical gap. It highlights the need for national-level guidelines on AI use, covering medico-legal accountability, local validation, and end-user training. Regulatory clarity will be essential to guide safe adoption. 

However, several limitations exist. The absolute sample size was small; the small sample reflects the limited national pool of radiologists with hands-on AI experience. Perspectives may differ among radiologists with limited or no exposure, who might exhibit greater apprehension or uncertainty toward AI adoption. Moreover, because participants were purposively sampled based on prior AI experience and willingness to participate, their views may reflect greater engagement or optimism toward AI. Radiologists less exposed to AI may emphasize different barriers or skepticism not captured here. Future studies should include such participants to capture the full spectrum of readiness. While UTAUT offers a strong structure for understanding technology adoption, it may overlook cultural and contextual variables like job security, institutional economics, and local healthcare workflows. Attitudes may also reflect perceptions of this specific AI tool, not AI in general. Comparative studies involving different AI platforms could clarify how algorithmic features influence clinician trust and workflow fit. The study’s modest sample size and reliance on self-reported data may introduce bias due to social desirability effects. However, participants represented a significant proportion of the eligible national pool with AI experience and accreditation, enhancing the study’s relevance. 

As aforementioned, the small sample reflects the limited national pool of radiologists with hands-on AI experience and positions this as an early-phase exploratory study generating hypotheses for subsequent large-scale evaluations of behavioral and workflow change. Future studies should triangulate attitudinal data with real-world workflow metrics, confidence calibration, and longitudinal assessments to examine how perceptions evolve with broader AI deployment.

## 5. Conclusions

In summary, AI adoption in breast cancer screening is not solely a question of technological capability but of alignment with clinical practice and radiologists’ psychological responses. Greater collaboration is needed between AI developers, healthcare institutions, and policymakers to overcome these barriers. This includes improving transparency in AI algorithms, offering ongoing training to reduce technological anxiety, and establishing robust ethical and legal frameworks that clearly define responsibility in AI-driven diagnostics.

## Figures and Tables

**Figure 1 cancers-17-03491-f001:**
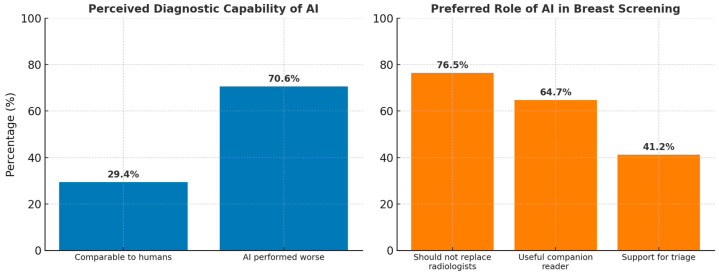
Radiologists’ perceptions of AI diagnostic capability and preferred roles in breast screening.

**Figure 2 cancers-17-03491-f002:**
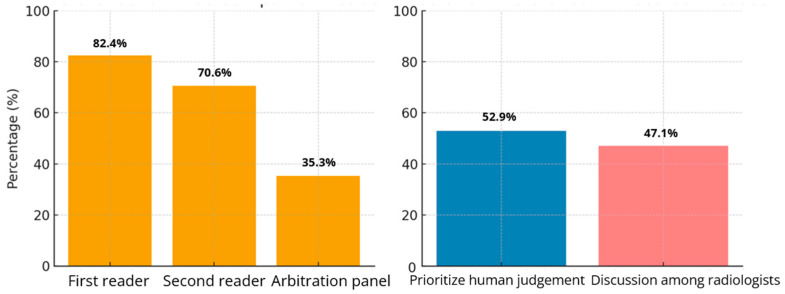
Radiologists’ Views on AI Use and Conflict Resolution in Screening Workflow, *n* = 17.

**Figure 3 cancers-17-03491-f003:**
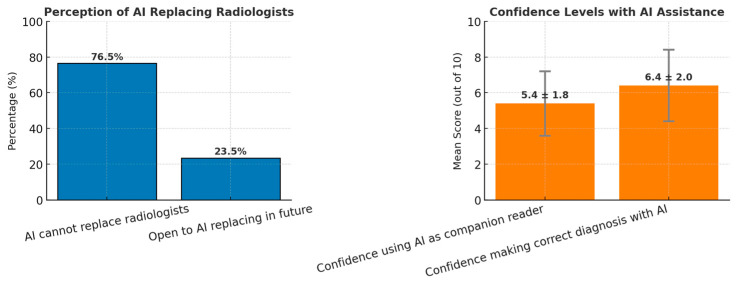
Radiologists’ perceptions and confidence in using AI in breast screening, *n* = 17.

**Figure 4 cancers-17-03491-f004:**
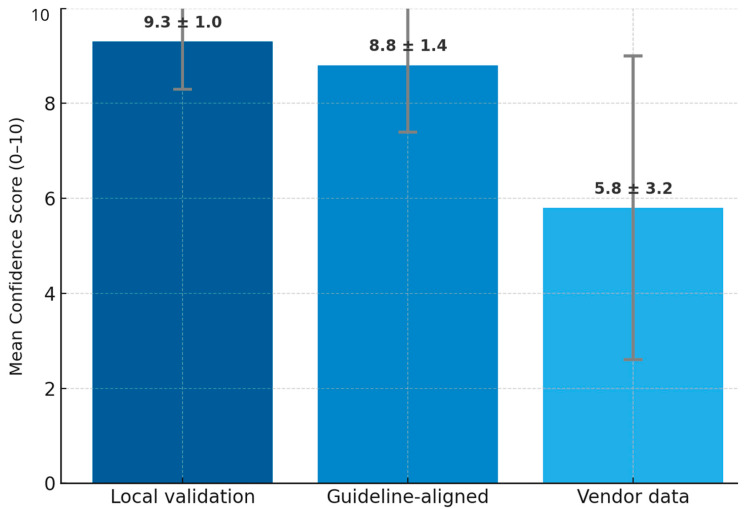
Bar graph showing radiologists’ confidence in AI based on source of validation, *n* = 17.

**Figure 5 cancers-17-03491-f005:**
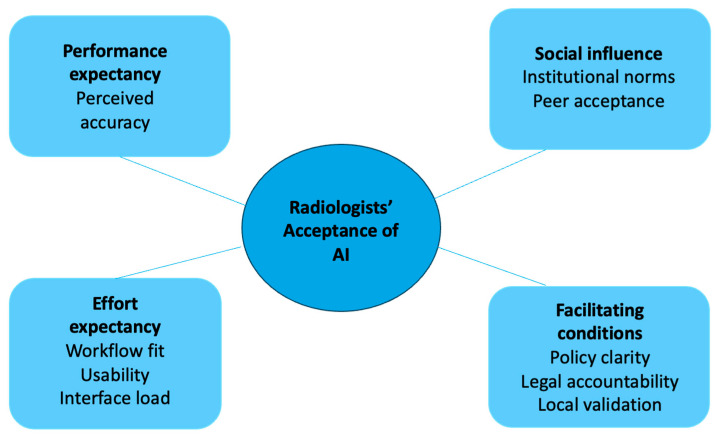
Thematic concept map summarizing key themes and findings after triangulation.

**Table 1 cancers-17-03491-t001:** Joint display of meta-inference of quantitative and qualitative study findings.

Quantitative Findings	Qualitative Themes and Subthemes ^1^	Meta-Inference
29.4% (5/17) of respondents rated AI as comparable to radiologists, while 70.6% (12/17) rated it as worse.	Theme 1: AI aids in routine diagnostic tasks but has limitations in complex interpretations. Subtheme 1.1: AI useful for routine tasks (Participant (P) P1, P2, P9) Subtheme 1.3: Limited in complex diagnostic integration (P4, P6)	Confirmation: The qualitative data aligns with survey responses, showing that AI is not yet viewed as capable of replacing human expertise for complex tasks. However, AI’s value lies in routine tasks and consistency.
76.5% (13/17) indicated that AI cannot replace human radiologists, but 23.5% (4/17) were open to replacement.	Theme 3: Ambivalent attitudes towards AI adoption. Subtheme 3.2: Fear and uncertainty about AI replacing humans (P2, P8).	Expansion: AI is considered a useful companion, though resistance remains to full replacement. There is also fear among radiologists about AI potentially replacing their role in the future.
Mean score of 6.4 (out of 10) for confidence in AI-assisted diagnosis.	Theme 3: Technological anxiety from AI use. Subtheme 3.3: Fear of over-reliance and doubts about vendor processes (P4, P5).	Discordance: While respondents expressed moderate confidence, qualitative insights reveal anxiety and resistance regarding over-reliance on AI tools.
64.7% (11/17) recommended AI as a companion for either Reader 1 or 2, while 58.8% (10/17) specifically preferred its use for Reader 2.	Theme 2: Integration and functional Alignment of AI in Clinical Workflows Subtheme 2.3: Integration barriers and infrastructure issues (P6, P10).	Expansion: The preference for AI use with secondary readers suggests it is better suited for supportive roles rather than primary diagnostic responsibility. Integration challenges must be addressed to enhance adoption.
Heat maps were ranked as the most useful feature, while triaging ranked fourth.	Theme 1: AI enhances detection in specific tasks. Subtheme 1.2: Heat maps improve cancer detection (P7).	Confirmation: There is alignment between the perceived utility of heat maps in the survey and their value in practice. AI’s value lies in augmenting human detection in nuanced areas.
52.9% (9/17) favored radiologists’ opinions prevailing over AI, and 47.1% (8/17) suggested discussion of discordant cases.	Theme 5: Increased workload from AI use. Subtheme 5.2: AI introduces workload paradox (P9).	Discordance: Although AI is intended to reduce workload, its use often increases work due to discordant case discussions and resulting additional investigations.
Testing AI on local datasets had a mean confidence rating of 9.3 (out of 10), the highest-rated evidence type.	Theme 4: Ethical and regulatory concerns. Subtheme 4.2: Importance of regulatory frameworks and national standards (P2, P3).	Expansion: Need for locally validated AI models and government regulations to foster AI adoption.

^1^ Theme = a major recurring idea across interviews; a subtheme = a smaller, related idea nested within that theme.

## Data Availability

Data available on reasonable request from the corresponding author.

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
