# Peer review of "Radiologists’ Perspectives on AI Integration in Mammographic Breast Cancer Screening: A Mixed Methods Study"

_cancers, 2025, doi:10.3390/cancers17213491_

Round 1
Reviewer 1 Report
Comments and Suggestions for Authors
The paper aims to evaluate the reception of AI tools in Singapore hospitals, providing first study of this kind for Asian population. While such research could be useful for AI software developers (even though the sample size is very limited - 10 / 17 subjects), it is presented in a very scattered way.
I think the quantitative results should be presented on charts rather than by numbers in text. One could then add also uncertainties to the calculated fractions. Maybe authors could also figure out some presentation method for the qualitative results, which would summarize the results "at first glance" - maybe some tree of related concepts?
I'd also welcome few words of introduction to the UTAUT approach, explaining why it is expected to work in this subject area. Also explain what authors mean by "coding" the model. Another question - why taking conclusions from quantitative and qualitative analysis is called "triangulation"? This has a quite specific meaning in image processing, but probably not the expected one.
It is also not quite clear who is the target reader of this manuscript. Software developers? Radiologists? I'd expect some more specific conclusions dedicated to such groups, for example what exactly is expected from software developers, possibly with some examples.
Finally, is everything OK with the sentence "64.7% (11/17) recommended AI for both first and second readers, while 58.8% (10/17) preferred AI for Reader 2."
Author Response
Comment 1: The paper aims to evaluate the reception of AI tools in Singapore hospitals, providing first study of this kind for Asian population. While such research could be useful for AI software developers (even though the sample size is very limited - 10 / 17 subjects), it is presented in a very scattered way. I think the quantitative results should be presented on charts rather than by numbers in text. One could then add also uncertainties to the calculated fractions. Maybe authors could also figure out some presentation method for the qualitative results, which would summarize the results "at first glance" - maybe some tree of related concepts?
Reply 1: We thank the reviewer for the comments and we agree that visualizing the quantitative results will improve comprehension. We have now created bar charts and confidence-interval plots (Figure 2-5) showing proportions with 95% CIs, where applicable. We have also added a thematic concept map (Figure 6) to summarize the key themes and concepts based on our findings and analysis.
Comment 2: I'd also welcome few words of introduction to the UTAUT approach, explaining why it is expected to work in this subject area. Also explain what authors mean by "coding" the model. Another question - why taking conclusions from quantitative and qualitative analysis is called "triangulation"? This has a quite specific meaning in image processing, but probably not the expected one.
Reply 2: Thank you for the comments. We have revised the Methods Subsection 2.3 to briefly explain that UTAUT helps explain technology acceptance through constructs of performance expectancy, effort expectancy, social influence, and facilitating conditions, which were deemed particularly relevant for understanding clinicians’ adoption of new digital tools. To clarify, “coding the model” refers to deductive coding using predefined UTAUT constructs as categories for qualitative analysis. We have re-phrased accordingly, "The analysis followed a deductive coding approach, where a codebook based on the UTAUT framework was developed prior to analyzing the transcripts. The codebook included predefined categories corresponding to the key constructs of the UTAUT model, such as performance expectancy, effort expectancy, social influence, and facilitating conditions. This theoretical framework guided the initial coding process. As the coding progressed, the categories were iteratively refined to capture emergent themes and nuances in the data that were not fully anticipated by the UTAUT model. Two authors (S.S.N.G. and Q.X.N.) independently generated codes in vivo by repeatedly engaging with the dataset. Any discrepancies between coding schemes were discussed and resolved collaboratively to achieve consensus."
To clarify as well, in mixed-methods research, “triangulation” refers to integrating multiple data sources to enhance validity, distinct from the image-processing use, we have further explained this in our manuscript.
Comment 3: It is also not quite clear who is the target reader of this manuscript. Software developers? Radiologists? I'd expect some more specific conclusions dedicated to such groups, for example what exactly is expected from software developers, possibly with some examples.
Reply 3: Thank you for the comment. We agree that the implications would differ by stakeholder and have added a new paragraph in the Discussion section to highlight more specific conclusions dedicated to each stakeholder group, "Arguably, the implications of these findings differ across stakeholder groups. For radiologists, the results emphasize the need for structured training programs and workflow integration strategies that enhance confidence in AI use while mitigating cognitive load and over-reliance. For developers, the emphasis lies in adopting human-centered design principles and ensuring rigorous validation of algorithms on local datasets to improve relevance, trust, and performance in specific population contexts. For policymakers and regulators, the study reinforces the importance of establishing clear medico-legal frameworks, transparent accountability structures, and harmonized national guidelines that define the clinical role of AI in breast cancer screening. All in all, stakeholder-specific strategies are critical for achieving safe, equitable, and sustainable integration of AI into real-world radiology practice."
Comment 4: Finally, is everything OK with the sentence "64.7% (11/17) recommended AI for both first and second readers, while 58.8% (10/17) preferred AI for Reader 2."
Reply 4: Thank you for spotting this. The second clause should read “while 58.8 % (10 / 17) supported its use as the secondary (Reader 2) companion”; the earlier value (64.7 %) already covers those who endorsed both readers. We have corrected this.
Reviewer 2 Report
Comments and Suggestions for Authors
This study addresses a highly relevant and timely topic, the perspectives of practicing radiologists on the adoption of AI in a critical screening domain. The use of a mixed-methods approach is appropriate and offers a richer, more nuanced understanding than a single method, effectively addressing complex technology adoption dynamics. The triangulation of findings is well-executed and clearly presented, particularly with the joint display and triangulation insights in the results section.
- What is the rational behind using unified theory of acceptance and use of technology framework instead of others framework? The authors should make this clearer in their method section.
- The study uses a concurrent, convergent mixed-methods design, as implied by the collection of both quantitative and qualitative data in a similar timeframe and their subsequent integration via meta-inference in a joint display (Table 1/Figure 1). While the implementation of the joint display and meta-inference is a strength, the design justification and naming could be more explicit in the Methods (Section 2.1 or 2.4). Clearly stating the specific mixed-methods design (e.g., convergent parallel design) and providing a clear rationale for why this particular design was chosen to address the research aims would strengthen the methodological rigor.
- The study focuses on radiologists in Singapore, which is a valuable contribution to the understanding of AI adoption in an Asian context. The quantitative sample size is small (N=17) (line 153), representing a specific, high-exposure group (at least 500 AI-assisted reads). This limits the generalizability of the survey findings to the broader population of screening radiologists, especially those with lower AI exposure or in different healthcare systems. This limitation is acknowledged (lines 316-317) but should be more prominently discussed in the context of the results. For example, how might the perspective of radiologists who have not done 500 AI-assisted reads differ?
- The qualitative sample of ten radiologists from all three national clusters is good, but the selection criteria (experience in utilizing AI and willingness to participate) suggests a sample potentially biased toward those more engaged with or affected by AI. Consider adding a brief discussion about how this might influence the qualitative themes observed.
- The questionnaire was based on a previous study [11] but was adapted to assess specific aspects (lines 92-95). For the adapted quantitative instrument, there's no mention of pilot testing or validation within this study's population. Brief evidence or a statement of the instrument's psychometric properties (e.g., reliability of the adapted scales) should be included or referenced if available, as adapting an instrument can impact its validity.
- Stating the quantitative survey had a 100% response rate (line 153) is excellent, but it should be clarified that this rate applies to the radiologists in the larger trial (N=17), not the entire population of eligible radiologists in Singapore.
- The UTAUT framework guided the qualitative analysis (lines 127-130). While the themes relate well to the framework, consider briefly stating how the core UTAUT constructs (e.g., performance expectancy, effort expectancy, social influence, and facilitating conditions) mapped to the emergent themes (e.g., Theme 1, 2, 4) in the Discussion, to explicitly leverage the theoretical contribution.
Author Response
Comment 1: What is the rational behind using unified theory of acceptance and use of technology framework instead of others framework? The authors should make this clearer in their method section.
Reply 1: Thank you for raising this point. We have now clarified in the Methods section that, "We selected the UTAUT framework because it is a parsimonious model that integrates constructs from eight earlier technology-adoption theories and it captures both behavioral intention and facilitating conditions, which the study team felt were particularly relevant for clinician technology adoption and hence suited to our study context [14]."
Comment 2: The study uses a concurrent, convergent mixed-methods design, as implied by the collection of both quantitative and qualitative data in a similar timeframe and their subsequent integration via meta-inference in a joint display (Table 1/Figure 1). While the implementation of the joint display and meta-inference is a strength, the design justification and naming could be more explicit in the Methods (Section 2.1 or 2.4). Clearly stating the specific mixed-methods design (e.g., convergent parallel design) and providing a clear rationale for why this particular design was chosen to address the research aims would strengthen the methodological rigor.
Reply 2: Thank you for the comments. We appreciate this observation. Indeed, the study employed a convergent parallel mixed-methods design, collecting quantitative and qualitative data concurrently and integrating findings via joint display meta-inference. We have now explicitly labelled and justified the design in the Methods section, "This study employed a convergent parallel mixed-methods design, in which quantitative and qualitative data were collected during overlapping periods and analyzed separately before being integrated through joint display meta-inference. This design was chosen to enable corroboration and complementarity between numerical trends and experiential insights on AI adoption.”
Comment 3: The study focuses on radiologists in Singapore, which is a valuable contribution to the understanding of AI adoption in an Asian context. The quantitative sample size is small (N=17) (line 153), representing a specific, high-exposure group (at least 500 AI-assisted reads). This limits the generalizability of the survey findings to the broader population of screening radiologists, especially those with lower AI exposure or in different healthcare systems. This limitation is acknowledged (lines 316-317) but should be more prominently discussed in the context of the results. For example, how might the perspective of radiologists who have not done 500 AI-assisted reads differ?
Reply 3: We acknowledge this limitation and agree that radiologists with limited AI exposure might perceive adoption differently, possibly with greater skepticism or uncertainty. We have now further elaborated upon this in our discussion of study limitations, "The absolute sample size was small; the small sample reflects the limited national pool of radiologists with hands-on AI experience. Perspectives may differ among radiologists with limited or no exposure, who might exhibit greater apprehension or uncertainty toward AI adoption. Moreover, because participants were purposively sampled based on prior AI experience and willingness to participate, their views may reflect greater engagement or optimism toward AI. Radiologists less exposed to AI may emphasize different barriers or skepticism not captured here. Future studies should include such participants to capture the full spectrum of readiness."
Comment 4: The qualitative sample of ten radiologists from all three national clusters is good, but the selection criteria (experience in utilizing AI and willingness to participate) suggests a sample potentially biased toward those more engaged with or affected by AI. Consider adding a brief discussion about how this might influence the qualitative themes observed.
Reply 4: We agree with the reviewer that self-selection of participants with AI experience could bias themes toward more nuanced or positive views of AI. We now discuss and acknowledge this in our discussion of study limitations, "Because participants were purposively sampled based on prior AI experience and willingness to participate, their views may reflect greater engagement or optimism toward AI. Radiologists less exposed to AI may emphasize different barriers or skepticism not captured here.”
Comment 5: The questionnaire was based on a previous study [11] but was adapted to assess specific aspects (lines 92-95). For the adapted quantitative instrument, there's no mention of pilot testing or validation within this study's population. Brief evidence or a statement of the instrument's psychometric properties (e.g., reliability of the adapted scales) should be included or referenced if available, as adapting an instrument can impact its validity.
Reply 5: We thank the reviewer for noting this. The adapted instrument was derived from de Vries et al. (2022) and reviewed for content validity by two senior breast radiologists for contextual relevance. Given the small sample and exploratory nature, formal psychometric validation was deemed not feasible and hence not undertaken. We have now clarified this in our Methods section.
Comment 6: Stating the quantitative survey had a 100% response rate (line 153) is excellent, but it should be clarified that this rate applies to the radiologists in the larger trial (N=17), not the entire population of eligible radiologists in Singapore.
Reply 6: We appreciate this clarification and have amended the sentence to specify that the 100% response rate refers to the radiologists participating in the multi-reader trial, not the entire national pool, "All 17 radiologists from the larger multi-reader trial completed the quantitative survey (100% response rate within this cohort)."
Comment 7: The UTAUT framework guided the qualitative analysis (lines 127-130). While the themes relate well to the framework, consider briefly stating how the core UTAUT constructs (e.g., performance expectancy, effort expectancy, social influence, and facilitating conditions) mapped to the emergent themes (e.g., Theme 1, 2, 4) in the Discussion, to explicitly leverage the theoretical contribution.
Reply 7: We agree that explicitly linking constructs to themes will strengthen the theoretical contribution. We have added a thematic concept map (Figure 6) and a concise mapping paragraph in the Discussion to address this, "The emergent themes from the interviews also align closely with the UTAUT constructs: Performance Expectancy corresponded to perceptions of AI’s accuracy and consistency (Theme 1); Effort Expectancy related to workflow fit and usability (Theme 2); Social In-fluence encompassed institutional culture and peer expectations (Theme 3); and Facilitating Conditions included regulatory clarity and local validation (Theme 4). This mapping underscores how the UTAUT model provides a coherent theoretical lens for understanding clinicians’ adoption behaviors (thematic concept map shown in Figure 6)."
Reviewer 3 Report
Comments and Suggestions for Authors
The article covers the important topic of acceptance of modern AI technologies by medical society. The problem is highly importaint and require the study of effects, but the research design looks a bit strange for me. Probably it requires the extension of survey. First - it covers only the human opinions, not tested alongside the real data or real hard cases. Second - it stops on the new topics generation with the open review and after the topics for research are found the review has ended. Thus - it has too small number of respondents (against the usual ~50) that covers only the first part of user behaviour study, typical for Product research. It does not have addmixing of data-based knowledge into the review and testing how sure the respondent in his opinion, it does not have the study of changes in the routine working procedure and does not cover the discussion of it with the new users, it does not cover the study of features of AI systems, only the overall acceptance or rejection based on the prior experinece in working with the certain system. It does not have also the base hypothesis we check in the statistical study. So I would suggest to extend the study further - it is highly important and provide the fruitful insights, but in this publication it looks like too much spaghetti-sliced to be the finished research on the acceptance and problems with the new widely-adopted product.
Author Response
Comment 1: The article covers the important topic of acceptance of modern AI technologies by medical society. The problem is highly importaint and require the study of effects, but the research design looks a bit strange for me. Probably it requires the extension of survey. First - it covers only the human opinions, not tested alongside the real data or real hard cases.
Reply 1: We thank the reviewer for this observation. The purpose of this study was not to evaluate algorithmic performance but to examine human–technology interaction and the contextual, behavioral, and organizational determinants of AI adoption in clinical screening. This complements, rather than duplicates, ongoing performance-validation studies already underway in Singapore (e.g., the multi-reader trial cited [13]). We have now clarified in the Methods section that, "This exploratory mixed-methods design was descriptive rather than inferential; no formal hypothesis testing was undertaken, and we sought to examine radiologists’ experiences and attitudes toward AI integration, complementing concurrent performance-validation trials in Singapore."
Comment 2: Second - it stops on the new topics generation with the open review and after the topics for research are found the review has ended. Thus - it has too small number of respondents (against the usual ~50) that covers only the first part of user behaviour study, typical for Product research. It does not have addmixing of data-based knowledge into the review and testing how sure the respondent in his opinion, it does not have the study of changes in the routine working procedure and does not cover the discussion of it with the new users, it does not cover the study of features of AI systems, only the overall acceptance or rejection based on the prior experinece in working with the certain system. It does not have also the base hypothesis we check in the statistical study. So I would suggest to extend the study further - it is highly important and provide the fruitful insights, but in this publication it looks like too much spaghetti-sliced to be the finished research on the acceptance and problems with the new widely-adopted product.
Reply 2: We agree that the small sample limits the generalizability and that this represents an exploratory phase of a broader implementation study. The sample size was ultimately constrained by the available national pool of radiologists with prior AI experience but still captures nearly the entire eligible population. To address this, we further emphasize in our discussion of study limitations that this is an early-phase exploratory investigation designed to generate hypotheses for larger follow-up studies. Given the small, purposive sample, this exploratory design was not powered for hypothesis testing but intended to describe attitudes and identify thematic patterns. We agree that future research should extend the sample size, include a broader spectrum of radiologists with varying levels of experience, and incorporate quantitative performance or workflow outcomes to complement perception data. We have thus noted these points in our Discussion section. Our aim was to capture radiologists’ perspectives before large-scale AI rollout, serving as a baseline for future longitudinal or interventional studies that will assess behavioral change after implementation, as a staged approach.
Round 2
Reviewer 2 Report
Comments and Suggestions for Authors
The authors have done a great job addressing the comments. However, I would suggest adding the percentage values on top of the bars in Figure 3 for improved clarity.
Author Response
Comment 1: The authors have done a great job addressing the comments. However, I would suggest adding the percentage values on top of the bars in Figure 3 for improved clarity.
Reply 1: Thank you for the kind words. We have now added percentage values directly above each bar in Figure 3 to enhance clarity and facilitate easier interpretation of the data.
Reviewer 3 Report
Comments and Suggestions for Authors
I still insist that, despite the importance of the research topic, the article's completeness limits the effect of its publication. I agree with the comments about the small number of users, which limits the possibility of conducting a comprehensive survey, and that testing the effectiveness of the algorithms is beyond the scope of this study. I still think that at least the current survey data should be presented in more clear way - in the less raw format and should provide some practical conclusions - what format of AI usage should be introduced to clinics, what percent of medical personel will benefit from it and what will see it as risk, some comparison of perceived risks with the real risks (on Figures). The quality and the content of all Figures in this research should be rewised. I suggest to remove unnecessary figures that can be replaced by simple text - i.e. Fig. 1 that contains mostly raw text.
Author Response
Comment 1: I still insist that, despite the importance of the research topic, the article's completeness limits the effect of its publication. I agree with the comments about the small number of users, which limits the possibility of conducting a comprehensive survey, and that testing the effectiveness of the algorithms is beyond the scope of this study. I still think that at least the current survey data should be presented in more clear way - in the less raw format and should provide some practical conclusions - what format of AI usage should be introduced to clinics, what percent of medical personel will benefit from it and what will see it as risk, some comparison of perceived risks with the real risks (on Figures).
Reply 1: We appreciate the reviewer’s continued concern. We have strengthened the limitations section by explicitly noting that the small sample size reflects the currently limited national pool of radiologists with real-world AI experience. We frame this study as exploratory early-adoption research and outline future plans for larger-scale multi-cluster evaluation as AI deployment expands.
We have also revisited the findings of the study and strengthened the practical implications of our discussion and conclusions, clarifying how perceived risks (e.g., workload increase due to false positives) are consistent with observed challenges in clinical use, "Notably, while 58.8% of participants in our study perceived that AI may increase workload due to the need to reconcile discordant findings (perceived risk), qualitative interviews confirmed this was already occurring in practice, with several radiologists describing additional cognitive and operational burden when AI flagged clinically non-actionable abnormalities (real risk). Our triangulated analysis demonstrates that radiologists’ concerns are grounded in concrete observations. What begins as a perceived risk (increased workload due to false-positive flags) is experienced as additional image review, supplementary investigations, and prolonged decision-making in discordant cases. This strongly emphasizes that the introduction of AI requires careful workflow redesign to ensure efficiency gains are realized rather than reversed."
Comment 2: The quality and the content of all Figures in this research should be rewised. I suggest to remove unnecessary figures that can be replaced by simple text - i.e. Fig. 1 that contains mostly raw text.
Reply 2: Thank you for the comments. We have re-examined the Figures and also improved the quality of the Figures where applicable. As suggested, the original Figure 1 contained mainly descriptive text and was therefore removed, with its content now summarized concisely in the Methods section. However, Figures presenting quantitative survey findings were retained because Reviewers 1 and 2 specifically requested more visual presentation of results rather than numerical reporting in text form. We seek your kind understanding in this matter.